# Deposition of Tetracoordinate Co(II) Complex with Chalcone Ligands on Graphene

**DOI:** 10.3390/molecules25215021

**Published:** 2020-10-29

**Authors:** Jakub Hrubý, Šárka Vavrečková, Lukáš Masaryk, Antonín Sojka, Jorge Navarro-Giraldo, Miroslav Bartoš, Radovan Herchel, Ján Moncol, Ivan Nemec, Petr Neugebauer

**Affiliations:** 1Central European Institute of Technology, CEITEC BUT, Purkyňova 656/123, 61200 Brno, Czech Republic; Jakub.Hruby@ceitec.vutbr.cz (J.H.); 200777@vutbr.cz (S.V.); Antonin.Sojka@ceitec.vutbr.cz (A.S.); Jorge.Navarro@ceitec.vutbr.cz (J.N.-G.); Miroslav.Bartos@ceitec.vutbr.cz (M.B.); Ivan.Nemec@ceitec.vutbr.cz (I.N.); 2Institute of Physical Engineering, Faculty of Mechanical Engineering, Brno University of Technology, Technická 2, 61669 Brno, Czech Republic; 3Department of Inorganic Chemistry, Faculty of Science, Palacký University, 17. listopadu 12, 77147 Olomouc, Czech Republic; lukas.masaryk01@upol.cz (L.M.); radovan.herchel@upol.cz (R.H.); 4Department of Inorganic Chemistry, Faculty of Chemical and Food Technology, Slovak University of Technology in Bratislava, 81237 Bratislava, Slovakia; jan.moncol@stuba.sk

**Keywords:** graphene, cobalt complexes, hybrid material, magneto-chemistry

## Abstract

Studying the properties of complex molecules on surfaces is still mostly an unexplored research area because the deposition of the metal complexes has many pitfalls. Herein, we probed the possibility to produce surface hybrids by depositing a Co(II)-based complex with chalcone ligands on chemical vapor deposition (CVD)-grown graphene by a wet-chemistry approach and by thermal sublimation under high vacuum_._ Samples were characterized by high-frequency electron spin resonance (HF-ESR), XPS, Raman spectroscopy, atomic force microscopy (AFM), and optical microscopy, supported with density functional theory (DFT) and complete active space self-consistent field (CASSCF)/N-electron valence second-order perturbation theory (NEVPT2) calculations. This compound’s rationale is its structure, with several aromatic rings for weak binding and possible favorable *π*–*π* stacking onto graphene. In contrast to expectations, we observed the formation of nanodroplets on graphene for a drop-cast sample and microcrystallites localized at grain boundaries and defects after thermal sublimation.

## 1. Introduction

Nearly three decades have already passed since the first description of the slow relaxation of magnetization in the polynuclear cluster [Mn_12_O_12_(O_2_CCH_3_)_16_(H_2_O)_4_] known as Mn_12_ [1,2,3], which started the whole new research field of molecular magnetism [4]. These so-called Single-Molecule Magnets (SMMs) exhibit magnetic bistability up to a specific blocking temperature manifested by an intrinsic spin-reversal barrier energy $\left(U_{\mathrm{eff}}\right)$. The barrier is a function of the total spin in the ground state (*S*) and the axial component of magnetic dipole–dipole interaction (*D*) as follows: $U_{\mathrm{eff}}=\left|D\right|\times S^{2}$ for integer spins and $U_{\mathrm{eff}}=\left|D\right|\times \left(S^{2}-\frac{1}{4}\right)$ for non-integer spins, respectively, in axial symmetry. This alone would imply that by increasing the number of magnetic centers, a better SMM would be obtained; however, there is a dependency of $D\propto \frac{1}{S^{2}}$ that stems from spin-orbit contributions to the *D*-tensor [5]. This dependency shifted the interest from rather large molecules with many magnetic atoms to Single-Ion Magnets (SIMs) [6]. Several challenges need to be addressed before fully utilizing these SIMs in real applications. One of the challenges is increasing the blocking temperature, which was recently found to be 80 K in dysprosium metallocene in 2018 [7]. This temperature, above the boiling point of liquid nitrogen (77 K), already holds promise for possible applications in spintronics [8], quantum computing [9], and molecular electronics [10]. The second challenge is finding the way from bulk material to functional surfaces.

The magnetic properties of magnetic molecules can be precisely measured by high-frequency electron spin resonance (HF-ESR) both in bulk [11,12,13,14,15,16,17,18,19,20] and on a surface [21,22,23]. Primarily, the Zeeman and zero-field-splitting (ZFS) contributions to the spin Hamiltonian with information about the intrinsic magnetic properties of a molecule can be determined. Today, the current effort is focused on making thin films, ordered arrays, or self-assembled monolayers that will lead to technological applications [24,25]. The key for this is to understand the behavior and adsorption of complex molecules on surfaces since their exposed surface offers many application possibilities but also brings many challenges, as these molecules can oxidize, decompose, or degrade in ambient conditions. There are two main ways to produce nanostructured magnetic thin films. They can be deposited onto a substrate via a wet-chemistry protocol from a solution [26,27,28] or by thermal sublimation in vacuum [29,30,31,32,33,34].

The electrical addressing of SMMs can be provided via a conductive substrate. A promising candidate is an atomically thin layer of graphite, known as graphene [35], which is an interesting substrate due to its high electron mobility [36,37], mechanical strength [38], and thermal conductivity [39]. The original preparation technique firstly used to prepare graphene in 2004 was micro-mechanical cleavage [40]. This method is suitable for tens-of-micrometers-large flakes; however, more industrial techniques for large and homogeneous surface coverage soon emerged, such as graphene production by chemical vapor deposition (CVD) [41], on silicon carbide [42], by liquid-phase exfoliation [43], or by large-scale roll-to-roll printing [44]. The perfect graphene is a zero-gap semiconductor, which helps the charge carrier mobility but also limits the applications.

Herein, we report on the synthesis, crystal structure, magnetic properties, and characterization of a new tetracoordinate complex [Co(4MeO-L)_2_Cl_2_] (**1**) with the chalcone imidazole-derivative ligand 4MeO-L = (2E)-3-[4-(1H-imidazol-1-yl)phenyl]-1-(4-methoxyphenyl)prop-2-en-1-on. The determination of the crystal structure revealed that this compound is tetracoordinate, and its molecules possess a unique shape with a large angle between the coordinated 4MeO-L ligands (vide infra). Tetracoordinate Co(II) compounds very often exhibit large easy-axis (*D* < 0) or easy-plane (*D* > 0) magnetic anisotropies [45]. Furthermore, the “flat” molecular shape involving the large aromatic system of the ligands might help to anchor complex molecules on surfaces such as graphene by non-covalent interactions. Therefore, we decided to thoroughly characterize the electronic structure of **1** by HF-ESR, to investigate both wet-chemistry and thermal sublimation depositions, and, thus, to produce a hybrid material composed of highly anisotropic Co(II)-based molecules and graphene. These samples were then characterized by Raman spectroscopy, X-ray photoelectron spectroscopy (XPS), and atomic force microscopy (AFM).

## 2. Results and Discussion

### 2.1. Synthesis and Crystal Structure of **1**

The chalcone ligand 4MeO-L was prepared by the aldol condensation of 4′-(imidazol-1-yl)benzaldehyde with 4-methoxyacetophenone, as is shown in Scheme 1. The purity and structure of 4MeO-L were confirmed by ^1^H and ^13^C NMR spectroscopy, mass spectrometry, and elemental analysis. The complex **1** was synthesized by a reaction between CoCl_2_·6H_2_O and 4MeO-L (molar ratio, 1:2) in methanolic solution, and it precipitated as a blue microcrystalline powder. Recrystallization from methanol led to the isolation of pale blue crystals suitable for single-crystal diffraction. The purity of **1** was confirmed by elemental analysis, and the phase purity, by powder diffraction experiments.

Compound **1** crystallizes in the monoclinic space group *Pc,* and it consists of tetracoordinate [Co(4MeO-L)_2_Cl_2_] molecules (Figure 1). The 4MeO-L ligands coordinate to the Co atom by the imidazolyl nitrogen atoms forming the Co-N bonds (*d*(Co-N) = 2.014(4) and 2.016(4) Å), while the chlorido ligands form longer bonds (*d*(Co-Cl) = 2.255(2) and 2.257(2) Å). The overall shape of the coordination polyhedron can be described best as a significantly distorted tetrahedron adopting *C*_2v_ pseudosymmetry (continuous shape measures index [46,47] for *T*_d_: 1.356 in **1**). The 4MeO-L ligands adopt an *E* conformation and remain planar even after coordination (Figure 1). The N-Co-N angle is wider than the Cl-Co-Cl one: <(N1-Co1-N2) = 125.7(2) vs. <(Cl1-Co1-Cl2) = 119.44(6).

The crystal packing in **1** is stabilized by a plethora of weak hydrogen bonds such as C-H···Cl and C-H···O. Remarkably, the large aromatic systems of the 4MeO-L ligands form π–π stacking interactions (the shortest C···C distances range between 3.27 and 3.45 Å), which stabilize the formation of supramolecular chains along the b crystallographic axis (Figure 1b).

### 2.2. Raman Vibrations

We used CVD graphene (Graphenea, San Sebastian, Spain) on a Si/SiO_2_ substrate. Figure 2 shows the substrate Raman spectra that helped us to determine the defects involved in the graphene. The Si/SiO_2_ Raman spectrum has a main strong phonon band at 520 cm^−1^ and two medium peaks at 301 cm^−1^ and in the region 946–976 cm^−1^ [48,49]. A spectrum of CVD graphene exhibited the strong peaks D at 1347 cm^−1^, G at 1595 cm^−1^, and 2D at 2689 cm^−1^, with the weaker peaks D′ at 1627 cm^−1^ and D+D′′ at 2462 cm^−1^. The presence of a strong G peak and weak D’ suggests CVD graphene with defects [50].

A comparison of bulk compound **1**, drop-cast, and two sublimated samples at 75 and 265 °C is illustrated in Figure 3. The Raman spectrum of the bulk compound **1** on the Si/SiO_2_ substrate consists of significant peaks (964, 1186, 1366, and 1603 cm^−1^) and peaks of Si/SiO_2_. In the case of the drop-cast sample, significant peaks were overlapped with the peaks of graphene and Si/SiO_2_, except one (1190 cm^−1^). By contrast, the Raman spectra of the sublimated samples all showed significant peaks due to measurements on a larger crystal and obtaining a stronger signal. The comparison tables of the Raman spectra can be found in Appendix A.

Optical images of the hybrid material taken along with Raman spectroscopy are shown in Figure 4. The molecules deposited by drop-casting formed small droplets up to 50 nanometers high. On the contrary, the molecules on sublimated samples formed microcrystals hundreds of nanometers high (see ESI, Appendix A).

### 2.3. Chemical Composition and Bonds

The chemical composition was probed by XPS. Figure 5 shows spectra of bulk compound **1** together with the molecular structure. 

The bulk compound **1** spectrum exhibited photoelectron peaks—Co 2p, Cl 2p, N 1s, C 1s, and O 1s—and Augers peaks: O_KLL_ and Co_LMM_. The detailed spectra of the selected peaks revealed specific chemical bonds. The N 1s peak was deconvoluted to two components: graphitic N with three neighboring C atoms and pyrrolic N with two C atoms and one Co bond [51]. The photoelectron peaks emitted from the p, d, and f electronic levels are further split by spin-orbit interactions. This helped us to distinguish, in the Cl 2p spectrum, between organic (Cl–C and Cl–H) and inorganic (Cl–Co) components [52]. Co 2p exhibited two main components and shake-up satellites. The spin-orbit shift of the main components Co 2p_3/2_ and Co 2p_1/2_ depends on the oxidation state, and with 15.6 eV, the Co(II) high-spin state predominates [53].

Figure 6 shows the comparison of the hybrid samples with CVD graphene: drop-cast, and sublimated at 75 °C and at 265 °C, respectively. In the drop-cast sample, we observed a decrease in the graphitic nitrogen component compared to bulk compound **1** and an apparent split of chlorine to inorganic and organic contributions. In the case of drop-casting, we detected a weak Co 2p signal on the surface, suggesting a possible complex decomposition (see ESI, Appendix A). In the case of the sublimated samples, even after several hours of acquisition, we did not obtain any convincing Co 2p peaks for 75 or 265 °C. This may be attributed to the possible partial decomposition of the complex or the surface sensitivity of XPS, with the complex outmost layers containing only a very few Co atoms or so-called “dead” layers with oxidized, spoiled molecules. This absence led us to a semi-empirical quantitative analysis of the powder after each sublimation (see ESI, Appendix A) and revealed an increased amount of cobalt and chlorine in the powder from the crucible compared to the bulk powder. This, along with the detected organic chlorine, suggests the possible partial chlorination of the graphene with a partial decomposition of the complex during both deposition processes. Carbon and oxygen contributions were discarded since they might be affected by adventitious contaminations due to the ex situ preparation procedures.

### 2.4. Molecular Adsorption by DFT

We investigated the adsorption of the molecule on graphene in the framework of density functional theory (DFT) using the Vienna Ab-Initio Simulation Package (VASP) [54,55,56,57]. The exchange–correlation potential was approximated by the generalized gradient approximation in Perdew–Burke–Ernzerhof (PBE) parametrization [58,59], the pseudopotential approach was used for the interaction between the valence electrons and ionic core, and Van der Waals forces were considered. Further details can be found in Section 3.

We performed a geometric relaxation calculation considering two possible geometries of the molecule relative to the graphene plane, as shown in Figure 7. The initial position of the molecule was chosen to mimic an AB-stacking configuration between the carbon rings of the molecule and the graphene substrate, as shown in Figure 7b,d, and also to take advantage of possible C–H···*π* hydrogen bonding between the hydrogen atoms of the molecule and *π* electrons of the carbon atoms in graphene. The molecule was placed manually on top of the substrate, such that the distances between the closest carbon atom of the molecule and the graphene plane was 3.13 and 3.20 Å for Geometries 1 and 2, respectively. During relaxation, the atoms of the molecule could move freely to their equilibrium positions, while the atoms of the substrate were kept fixed.

After relaxation, we found that the molecule bound to the substrate in both configurations, with distances of 3.31 Å (Geometry 1) and 3.29 Å (Geometry 2) between the closest carbon atom of the molecule and the graphene plane. Such distances correspond to the typical distances between *π–π*-bonded carbon rings, and we found that there was no considerable change in the molecular shapes after the adsorption. The binding energies were 0.89 eV per molecule (85.4 kJ/mol) for Geometry 1, and 1.08 eV per molecule (104.0 kJ/mol) for Geometry 2, where the main contribution to this energy comes from the van der Waals interactions between the carbon atoms of the molecule and substrate. If van der Waals forces are not considered, the binding energies fall to the meV range, below the thermal energy at room temperature (25.8 meV at 300 K). Therefore, van der Waals forces play a crucial role in the adsorption of these cobalt-based molecules on graphene.

### 2.5. HF-ESR Spectroscopy

Figure 8 shows the HF-ESR spectra acquired for bulk compound **1** at four frequencies—380, 415, 456, and 490 GHz—while sweeping the magnetic field from 0 to 15 T at 5 K.

The used effective spin Hamiltonian for the simulations is the following Equation (1):(1)$$\hat{H}={\hat{H}}_{\mathrm{Zeeman}}+{\hat{H}}_{\mathrm{ZFS}}=\mu_{B}B\cdot g\cdot \hat{S}+D\left[{\hat{S}}_{z}^{2}-\frac{1}{3}S\left(S+1\right)\right]+E\left({\hat{S}}_{x}^{2}-{\hat{S}}_{y}^{2}\right)$$
where $\mu_{B}$ is the Bohr magneton, ***B*** is the external magnetic field, g is a tensor linking the external magnetic field with spin vectors, $\hat{S}$ is the electron spin operator, and *D* and *E* are axial and rhombic zero-field splitting parameters, respectively. The best fit was found for the spin Hamiltonian parameters as follows: *D* = 14.6 cm^−1^ with *E/D* = 0.235, and *g*_x_ = 2.32, *g*_y_ = 2.38, and *g*_z_ = 2.16 (Table 1).

### 2.6. CASSCF Calculations

To support the analysis of the HF-ESR spectra of **1**, we performed complete active space self-consistent field (CASSCF) calculations complemented by N-electron valence second-order perturbation theory (NEVPT2) using an ORCA 4.2 computational package [60]. The details of the calculations are explained in Section 3—*Theoretical Methods*. The spin Hamiltonian parameters were extracted by utilizing the effective Hamiltonian theory and we obtained a set of the ZFS parameters—for *S* = 3/2, *D* = + 14.5 cm^−1^ and *E*/*D* = 0.15—and the anisotropy of the *g*-tensor components was confirmed (*g*_x_ = 2.325, *g*_y_ = 2.378, *g*_z_ = 2.163, and *g*_av_ = 2.289). These values are in good agreement with the values obtained by HF-ESR spectroscopy. Next, we performed additional calculations for the optimized Geometries 1 and 2 of the [Co(4MeO-L)_2_Cl_2_] molecules deposited on the graphene surface as calculated by periodic DFT. The resulting ZFS parameters are, besides the slightly lower rhombicity, rather similar to those calculated for **1** (all the calculated values are summarized in Table 1). The visualizations of the calculated *D*-tensor principal axes overlaid over the structures of the complex molecules (Figure 9) underline the similarities among **1** and Geometries 1 and 2. The directions of *D*_Z_ are practically the same in all the studied molecules. However, the directions of the *D*_X_ and *D*_Y_ axes differ among the studied molecules (Figure 9).

## 3. Materials and Methods 

### 3.1. Materials 

CoCl_2_·6H_2_O was bought from PMRLab (Port Elizabeth, South Africa), and 4-(1H-imidazol-1-yl)benzaldehyde, 4-methoxyacetophenone, NaOH, and the solvents (methanol (MeOH), diethyl ether (Et_2_O), and the deuterated solvents for the NMR experiments (deuterated chloroform (CDCl_3_))) were purchased from VWR International (Stříbrná Skalice, Czech Republic), Sigma-Aldrich (Prague, Czech Republic), Lach-Ner (Neratovice, Czech Republic), and Litolab (Chudobín, Czech Republic). 

### 3.2. Synthesis

#### 3.2.1. (2*E*)-3-[4-(1H-imidazol-1-yl)phenyl]-1-(4-methoxyphenyl)prop-2-en-1-one (4MeO-L)

A methanolic sodium hydroxide solution (40%; 1.2 mL) was added dropwise to a mixture of 4-methoxybenzaldehyde (2 mmol, 0.300 g), 4’-(imidazol-1-yl)benzaldehyde (2 mmol, 0.377 g), and methanol (5 mL) over a period of 40 min. The resulting solution was stirred at room temperature until the completion of the reaction. The precipitate was filtered off and washed with a cold methanol–water mixture (1:10). The resulting product was recrystallized from methanol and dried in a desiccator under reduced pressure (overnight) [61].

Yellowish solid. Yield: 83%. ^1^H NMR (400 MHz, Chloroform-*d,* 298 K, ppm) δ 8.06 (d, *J* = 8.4 Hz, 2H, C17-H, C21-H), 7.92 (s, 1H, C2-H), 7.84–7.74 (m, 3H, C8-H, C10-H, C13-H), 7.57 (d, *J* = 15.6 Hz, 1H, C12-H), 7.46 (d, *J* = 8.1 Hz, 2H, C7-H, C11-H), 7.34 (s, 1H, C5-H), 7.24 (s, 1H, C4-H), 7.00 (d, *J* = 8.4 Hz, 2H, C18-H, C20-H), 3.91 (s, 3H, C23-H). ^13^C NMR (101 MHz, Chloroform-*d,* 298 K, ppm) δ 188.28 (C14), 163.59 (C19), 142.12 (C12-H), 138.43 (C6), 135.38 (C2-H), 134.23 (C16), 132.72 (C9), 130.87 (C4-H), 130.85 (C8-H, C10-H), 129.86 (C17-H, C21-H), 122.51 (C13-H), 121.45 (C7-H, C11-H), 117.84 (C5-H), 113.91 (C18-H, C20-H), 55.52 (C23-H). ESI+MS (MeOH, *m*/*z*): 305.27 (calc. 305.12; 100%; {4MeO-L + H}^+^), 327.11 (calc. 327.11; 79%; {4MeO-L + Na}^+^), 630.81 (calc. 631.67; 88%; {2 × 4MeO-L + Na}^+^). IR (ATR, v, cm^−1^): 407w, 447w, 521w, 593w, 653w, 770w, 816s, 905w, 958w, 981w, 1015m, 1061w, 1120w, 1168m, 1225m, 1254w, 1309w, 1342w, 1433w, 1523s, 1588s, 1658m, 3103w.

#### 3.2.2. Complex [Co(4MeO-L)_2_(Cl)_2_] (**1**)

The solution of CoCl_2_^.^6H_2_O (1 mmol, 0.237 g) in 5 mL of methanol was heated up to 50 °C, and then, 2 molar equiv. of 4MeO-L was added (2 mmol, 0.608 g). The solution was cooled down and stirred at ambient temperature for 2 h. The obtained blue precipitate was collected by filtration and washed with water (2 × 0.5 mL) and Et_2_O (2 × 1 mL). The blue solid product was dried in a desiccator under reduced pressure (overnight) [62].

Blue solid. Yield: 92%. Anal. Calc. for CoC_38_H_32_Cl_2_N_4_O_4_ (**1**): C, 61.80; H, 4.37; N, 7.59%; found: C, 61.59; H, 4.31; N, 7.42%. ESI+MS (MeOH, *m*/*z*,): 305.34 (calc. 305.7; 10%; {4MeO-L + H}^+^), 471.07 (calc. 471.21; 100%; {[Co(4MeO-L)(Cl)_2_] + 2H_2_O + H}^+^), 702.21 (calc. 702.14; 71%; {[Co(4MeO-L)_2_(Cl)]}^+^), 774.76 (calc. 775.13; 30%; {[Co(4MeO-L)_2_(Cl)] + 4H_2_O}^+^). IR (ATR, v, cm^−1^): 399w, 412w, 476w, 501w, 517w, 589w, 612w, 646w, 729w, 809s, 969w, 1013w, 1059w, 1102w 1134w, 1159w, 1212w, 1347w, 1401w, 1497w, 1534w, 1598s, 3129w. Thermal stability up to ca. 310 °C was confirmed by thermogravimetry measurement.

### 3.3. Deposited Samples

Drop-cast sample was prepared by dissolving the bulk compound **1** in dichloromethane (98%, Penta, Czech Republic) to make a final solution with a 100 μM concentration. The actual drop-casting was conducted in a mobile glove bag (Merck, Germany) filled with inert nitrogen gas; 10 μL was drop-cast onto a substrate. For the thermal sublimation, we used a home-built high-vacuum sublimation chamber equipped with a quartz crucible heated by tungsten wire, with a thermocouple in thermal contact with the crucible. The base chamber pressure during the sublimation was 2 × 10^−7^ mbar. The sublimations were performed at 75 and 265 °C, respectively. 

### 3.4. Raman Spectroscopy (RS)

Raman spectra were acquired on a confocal Raman microscope WITec Alpha300 R+ (WITec, Ulm, Germany). All measurements were carried out with the excitation laser source with a 532 nm wavelength and 1 mW power output. Optical images were acquired with a 100× objective (NA 0.9, WD 0.31 mm).

### 3.5. Atomic Force Microscopy (AFM)

All topography images and profiles were obtained with the scanning probe microscope Bruker Dimension Icon in tapping mode.

### 3.6. X-ray Photoelectron Spectroscopy (XPS)

X-ray photoelectron (XPS) measurements were carried out with a Kratos Axis Supra (Kratos Analytical, Manchester, United Kingdom) spectrometer at room temperature and under ultra-high vacuum (UHV) conditions. The instrument was equipped with a monochromatic Al Kα source of 1486.6 eV (15 mA, 15 kV) and a hemispherical analyzer with a hybrid magnetic and electrostatic lens for enhanced electron collection. Survey and detailed XPS spectra were acquired at normal emission with fixed pass energies of 160 and 20 eV, respectively. All spectra were calibrated to the hydrocarbon peak set to 284.8 eV. The Kratos charge neutralizer system was used on all specimens. The inelastic backgrounds in all the spectra were subtracted according to the Shirley method [63]. Data analysis was based on a standard deconvolution method using a mixed Gaussian (G) and Lorentzian (L) line shape (G = 70% and L = 30%, Gaussian–Lorentzian product) for each component in the spectra. The elemental composition of the samples was evaluated using a semi-empirical approach. The integrated intensity of each component was corrected with the photoionization cross-section calculated for each atom, neglecting the differences in photoelectron escape length as a function of the kinetic energy [64]. The spectra were analyzed using the CasaXPS software (version 2.3.18).

### 3.7. High-Frequency Electron Spin Resonance (HF-ESR)

HF-ESR spectra were acquired on a newly home-built spectrometer featuring a signal generator (Virginia Diodes, Charlottesville, VA, USA), an amplifier–multiplier chain (Virginia Diodes, Charlottesville, USA), a quasi-optical bridge (Thomas Keating, Billingshurst, UK), and a 16 T solenoid cryomagnet (Cryogenic, London, UK) with heterodyne signal detection. The reference powder sample of the complex was studied as a pressed powder with a ø 5 mm pellet sample. All ESR spectra were simulated using EasySpin [65], a toolbox for Matlab.

### 3.8. Density Functional Theory (DFT)

The density functional calculations for molecular adsorption were performed with the Vienna Ab-Initio Simulation Package (VASP) [54,55,56,57] version 5.4.4, which uses a plane-wave basis for the Kohn–Sham orbitals, the Projector Augmented Wave (PAW) method [57,66], and pseudopotentials. The exchange–correlation potential was approximated by generalized gradient approximation in Perdew–Burke–Ernzerhof (PBE) parametrization [58,59]. Van der Waals corrections were calculated using the D2 method of Grimme [67]. In all calculations, the kinetic energy cut-off for the plane waves was 420 eV. For the calculation of the ground state energy of the system molecule+substrate and graphene substrate, a Γ-centered 2 × 2 × 1 Monkhorst–Pack mesh [68] was used to sample the Brillouin zone, while a Γ-point calculation was used for the ground-state energy of the molecule. We considered two different molecular geometries relative to the graphene plane. Geometry 1 lies on top of a 17 × 8 graphene supercell, while a 13 × 7 supercell was used for Geometry 2 (1 × 1 corresponds to graphene’s unit cell). Since a plane-wave basis was used, the systems were periodic along each lattice vector; therefore, an array of infinite molecules was simulated, which in principle can interact with each other. Nevertheless, the distance between the closest atoms of neighboring molecules was no less than 9.0 Å for Geometry 1 and 5.8 Å for Geometry 2 (the distance between the closest Co atoms was 17.2 Å for Geometry 1, and 19.7 Å for Geometry 2); therefore, it was assumed that the molecules did not interact with each other. Geometry relaxation was performed until the forces were below 0.1 eV/Å. 

### 3.9. Theoretical Methods (CASSCF/NEVPT2)

All the theoretical calculations were performed with the ORCA 4.2 computational package [69]. All the calculations employed the triple-ζ def2-TZVP basis functions for all atoms except for carbon and hydrogen, for which the def2-SVP basis set was applied [70]. Additionally, the def2/J and def2-TZVP/C auxiliary basis sets were utilized together with RIJCOSX approximation [71,72]. The multiconfigurational character of the studied Co(II) complexes was handled by calculations utilizing self-consistent field (SA-CASSCF) wave functions [73] with N-electron valence second-order perturbation theory (NEVPT2) [74]. The active space of the CASSCF calculation was set to five d-orbitals of Co(II) (CAS(7,5)). The *D*- and *g-*tensors, based on dominant spin−orbit coupling contributions from excited states, were calculated through quasi-degenerate perturbation theory (QDPT) [75]. We utilized approximations to the Breit–Pauli form of the spin-orbit coupling operator (SOMF approximation) [76] and effective Hamiltonian theory [77]. 

### 3.10. Elemental Analyses (EA)

Elemental analysis was carried out using a Flash 2000 CHNS Elemental Analyzer (Thermo Scientific, Waltham, MA, USA).

### 3.11. Mass Spectrometry (MS)

Electrospray ionization mass spectrometry (ESI-MS; methanol solutions) was performed with an LCQ Fleet ion trap spectrometer (Thermo Scientific, Waltham, MA, USA; QualBrowser software, version 2.0.7) in both positive (ESI+) and negative (ESI-) ionization modes.

### 3.12. NMR Spectroscopy

^1^H and ^13^C NMR spectroscopy, and ^1^H-^13^C gsHMQC and ^1^H-^13^C gsHMBC two-dimensional correlation experiments were performed using CDCl_3_ (4MeO-L) solution at 300 K using a Varian spectrometer (Palo Alto, CA, USA) at 400.00 MHz (for ^1^H NMR) and 101.00 MHz (for ^13^C NMR), where gs = the gradient selected, HMQC = the heteronuclear multiple quantum coherence, and HMBC = the heteronuclear multiple bond coherence. ^1^H and ^13^C NMR spectra were calibrated against the residual CDCl_3_^1^H NMR (7.26 ppm) and ^13^C NMR (77.16 ppm) signals. The splitting of the proton resonances in the reported ^1^H spectra is defined as s = singlet, d = doublet, dd = doublet of doublets, sep = septet, m = multiplet, and bs = broad signal.

### 3.13. Infrared Spectroscopy

A Jasco FT/IR-4700 spectrometer (Jasco, Easton, MD, USA) was used for the collection of the infrared (IR) spectra of the studied ligand and complex in the range of 400–4000 cm^−1^ by using the attenuated total reflection (ATR) technique on a diamond plate.

### 3.14. Crystallography

A single crystal of **1** was mounted on a Stoe StadiVari diffractometer possessing a Pilatus3R 300 K detector and microfocused sealed tube Xenocs Genix3D Cu HF (*λ* = 1.54186 Å) at 100 K. The structure was solved using the program SuperFlip [78] and refined using the program ShelXL (ver. 2018/3) [79] in the crystallographic package Olex2 [80]. The structure was drawn using the Mercury program [81]. Crystal data for CoC_38_H_32_Cl_2_N_4_O_4_ (*M* = 738.50 g/mol): monoclinic, space group *Pc* (no. 7), *a* = 18.7700(3) Å, *b* = 12.2910(4) Å, *c* = 7.3969(6) Å, *β* = 101.392(3)°, *V* = 1672.86(15) Å^3^, *Z* = 2, *T* = 100(1) K, *μ*(CuKα) = 5.885 mm^−1^, *D*_calc_ = 1.466 g/cm^3^, 31,686 reflections measured (3.596° ≤ 2Θ ≤ 72.338°), 5367 unique (*R*_int_ = 0.0372), used in all calculations. The final R1 was 0.0455 (I > 2σ(I)), and the wR_2_ was 0.1211 (all data). The highest peak: +0.28; the deepest hole: −0.53. Crystal structure refinement: All atoms except for hydrogen were refined anisotropically. The hydrogen atoms were placed into the calculated positions, and they were included into the riding-model approximation with *U*_iso_ = 1.2*U*_eq_(C) or 1.5 *U*_eq_ (CH_3_) and d(C−H) = 0.95–0.98 Å.

## 4. Conclusions

This paper reports on the synthesis, crystal structure, magnetic properties, and characterization of a new Co(II)-based complex with monodentate chalcone ligands and its deposition on graphene. The magnetic properties were determined from HF-ESR measurements and were found to be in fair agreement with CASSCF/NEVPT2 ab initio quantum chemical calculations. The spin Hamiltonian parameters are as follows: *D* = 14.6 cm^−1^ with significant rhombicity *E/D* = 0.235, and *g*_x_ = 2.32, *g*_y_ = 2.38, and *g*_z_ = 2.16. Depositions on graphene were attempted by both drop-casting in an inert nitrogen atmosphere and by the thermal sublimation of bulk compound **1** in a high vacuum. In both cases, we observed organic chlorine components, suggesting the partial decomposition of the complex or possible chlorination of graphene. On the contrary, the Raman spectra showed a good agreement of the peaks in bulk and on the graphene; however, a few peaks from the complex overlapped with the graphene peaks, which hindered the analysis. In the case of the drop-cast sample, we observed the formation of small nanodroplets about 50 nm high on the graphene. Samples prepared by thermal sublimations revealed the formation of microcrystallites formed mostly at the grain edges and defects on graphene. DFT simulations of the complex at two geometries on the graphene surface confirmed only weak attraction to the graphene surface, with the crucial role of van der Waals forces in the adsorption on graphene. The outlook for the successful deposition of intact complexes on graphene surfaces requires the fine chemical tailoring of ligands, promoting adhesion on graphene, and utilizing chelation agents that protect the complex from detrimental effects such as atmospheric moisture, oxidation, and thermal decomposition. The next step after successful deposition is to obtain the magnetic properties of a thin film on the surface, which will be obtained from HF-ESR measurements or from X-ray magnetic circular dichroism (XMCD) at the synchrotron facility.

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
