# Peer review of "Deposition of Tetracoordinate Co(II) Complex with Chalcone Ligands on Graphene"

_molecules, 2020, doi:10.3390/molecules25215021_

Round 1
Reviewer 1 Report
The manuscript under review describes an interesting phenomenon of formation of nanodroplets of a Co(II)-based complex with chalcone ligands on graphene and microcrystallites at grain boundaries including defects after thermal sublimation.
The synthesized samples were characterized in details by HF-ESR, XPS, Raman, AFM, optical microscopy and also the DFT and CASCCF/NEVPT2 calculations were carried out. I like a high quality of the results presentation.
My only comment concerns just the last point. Using the VASP package the authors have performed periodic DFT calculations in Gamma point. However the system studied has a pronounced planar geometry. Thus using 2x2 k-point grid in graphene plane would be more relevant (and accurate) in this case.
Author Response
Manuscript ID: molecules-973716
We thank for reviewer comment. The changes to submitted manuscript are listed below:
Reviewer #1 commented on paper: “The manuscript under review describes an interesting phenomenon of formation of nanodroplets of a Co(II)-based complex with chalcone ligands on graphene and microcrystallites at grain boundaries including defects after thermal sublimation.
The synthesized samples were characterized in details by HF-ESR, XPS, Raman, AFM, optical microscopy and also the DFT and CASCCF/NEVPT2 calculations were carried out. I like a high quality of the results presentation. “
Reviewer#1/Question1: My only comment concerns just the last point. Using the VASP package the authors have performed periodic DFT calculations in Gamma point. However the system studied has a pronounced planar geometry. Thus using 2x2 k-point grid in graphene plane would be more relevant (and accurate) in this case.
Answer to Reviewer#1/Question1: We appreciate the reviewer’s comment and agree with it. For the calculation of the ground-state energy of the system Molecule-Substrate, and of the graphene substrate without molecule, we have increased the k-point grid to a 2x2x1 Monkhorst-Pack grid, centered at the Gamma point. In this sense, the Brillouin zone was sampled by 4 vectors, which included the Gamma point. Nevertheless, to calculate the ground-state energy of the molecule without substrate, we kept a Gamma point calculation. We have found that the distance of the closest carbon atom of the molecule to the graphene plane went from 3.24 Å to 3.31 Å in geometry 1, while it remained at 3.29 Å in geometry 2. The adsorption energies also changed, going from 0.77 eV/Å to 0.89 eV/Å in geometry 1, and from 1.00 eV/Å to 1.08 eV/Å in geometry 2.
The changes in the main text are as follows:
(Lines 182-184) After relaxation, we found that the molecule binds to the substrate in both configurations, with a distance of 3.31 Å (geometry 1) and 3.29 Å (geometry 2) between the closest carbon atom of the molecule to the graphene plane.
(Lines 186-187) The binding energies were 0.89 eV per molecule (85.4 kJ/mol) for geometry 1, and 1.08 eV per molecule (104.0 kJ/mol) for geometry 2,
The methods section was also updated, and now reads:
(Lines 310-312) For the calculation of the ground state energy of the system molecule+substrate and graphene substrate, a Γ-centered 2x2x1 Monkhorst-Pack mesh [67] was used to sample the Brillouin zone, while a Γ-point calculation was used for the ground-state energy of the molecule.
With the revisions presented above, we gratefully thank for the reviewer suggestion and we are convinced that the manuscript is improved and ready to be published in MDPI Molecules.
On behalf of all authors, yours faithfully,
Jakub Hrubý and Petr Neugebauer
Reviewer 2 Report
The manuscript “Deposition of tetracoordinate Co(II) complex with chalcone ligands on graphene” is interesting and worth of publication after revisions.
I have only some question and remarks.
1) what do author mean with "bulk powder"? Is it only metal complex or does it contain CVD graphene? If it is only metal complex, why bulk raman spectra contains Si/SiO2 peaks? If bulk powder contains graphene, what does "Drop-cast sample was prepared by dissolving the bulk powder complex in dichloromethane" mean?
2)What are Raman peaks at about 22680 cm-1 for Drop-cast and sublimated samples? I haven’t found any mention for that in the text.
3)Description of how the starting geometries for DFT calculations have been generated is not exhaustive. Has the complex been placed manually or by some kind of docking procedure?
4)In general, for each used technique a more detailed description of the sample investigated (metal complex or hybrid material) should be given also in the result and discussion section
5) conclusion section: In addition to conclusions about deposition procedures and results, are speculations about magnetic properties of these hybrid compounds possible?
Minor remarks:
Line 96 “….plethora of weak hydrogen bonds such as C-H···O and C-H···O.” Please check
Line 347 “monolinic”. Please correct
Author Response
Manuscript ID: molecules-973716
We thank for reviewer comments and hints for our paper. The changes to submitted manuscript are listed below:
Reviewer #2 commented on paper: “The manuscript “Deposition of tetracoordinate Co(II) complex with chalcone ligands on graphene” is interesting and worth of publication after revisions. I have only some question and remarks.
Reviewer#2/Question1: What do author mean with "bulk powder"? Is it only metal complex or does it contain CVD graphene? If it is only metal complex, why bulk raman spectra contains Si/SiO2 peaks? If bulk powder contains graphene, what does "Drop-cast sample was prepared by dissolving the bulk powder complex in dichloromethane" mean?
Answer to Reviewer#2/Question1: We agree that bulk powder might have ambiguous meaning, it is as-synthesized metal complex, it does not contain any CVD graphene, bulk raman spectra contain Si/SiO2 peak because this substrate was used as a base for metal complex powder, therefore we added to the manuscript:
Line 117: “bulk compound 1”
Line 118: “bulk compound 1 on Si/SiO2 substrate”
Line 126: “bulk compound 1” in the Figure 3 description
Line 136: “bulk compound 1”
Line 150: “compound 1”
Reviewer#2/Question2: What are Raman peaks at about 22680 cm-1 for Drop-cast and sublimated samples? I haven’t found any mention for that in the text.
Answer to Reviewer#2/Question2: We believe that reviewer meant peaks around 2680 cm-1 for drop-cast and sublimated samples that denote 2D graphene peak in our sample and they are compared for specific samples in Table S1.
Reviewer#2/Question3: Description of how the starting geometries for DFT calculations have been generated is not exhaustive. Has the complex been placed manually or by some kind of docking procedure?
Answer to Reviewer#2/Question3: We appreciate the reviewer’s comment and agree that the choice of the starting geometries need to be discussed in the text. To answer to the comment, we have used following geometries (see below) since they take advantage of the AB stacking between carbon rings of the molecule and graphene, as well as possible Hydrogen bonding between the hydrogen atoms of the molecule and the carbon atoms in (their π electrons) graphene. The initial positioning of the molecule was done manually to the mentioned AB-stacking, as observed in Fig. 7 (b,d). As mentioned in the main text, the initial position of the closest carbon atom of the molecule to the graphene plane was 3.13 Å and 3.20 Å for geometries 1 and 2, respectively. During relaxation, all the atoms of the molecule could freely move to their equilibrium position, while the atoms in graphene were kept fixed.
To clarify these considerations, we have included the following in the main text:
(lines 175-179) “The initial position of the molecule was chosen to mimic an AB-stacking configuration between the carbon rings of the molecule and the graphene substrate, as shown in Figure 7 (b, d), and also to take advantage of possible C-H···π hydrogen bonding between the hydrogen atoms of the molecule and π electrons of carbon atoms in graphene. The molecule was placed manually on top of the substrate,…”
(Lines 181-182) “During relaxation, the atoms of the molecule could move freely to their equilibrium positions, while the atoms of the substrate were kept fixed.”
Reviewer#2/Question4: In general, for each used technique a more detailed description of the sample investigated (metal complex or hybrid material) should be given also in the result and discussion section
Answer to Reviewer#2/Question4: We agree that on some places it should be clearer what type of sample was measured, therefore, we added to the main text:
Line 109: “substrate”
Line 128: “for hybrid material”
Line 189: “of bulk compound 1”
Line 268: “compound 1”
Line 373: “compound 1”
Reviewer#2/Question5: Conclusion section: In addition to conclusions about deposition procedures and results, are speculations about magnetic properties of these hybrid compounds possible?
Answer to Reviewer#2/Question5: We added a sentence to conclusion with outlook on magnetic measurements, that are possible either by HF-ESR which we plan to conduct or by XMCD on synchrotron, which is more demanding: “The next step after successful deposition is to obtain magnetic properties of a thin film on surface that will be obtained from HF-ESR measurements or from X-ray magnetic circular dichroism (XMCD) at synchrotron facility.”
Minor remarks:
Comment: Line 96 “….plethora of weak hydrogen bonds such as C-H···O and C-H···O.” Please check
Thank you for spotting these typos:
- Change on Line 96:
OLD: “C-H···O and C-H···O”
NEW: “C-H···Cl and C-H···O”
Comment: Line 347 “monolinic”. Please correct
- Change on Line 347:
OLD: “monolinic”
NEW: “monoclinic”
With the revisions presented above, we gratefully thank for the reviewer suggestions and we are convinced that the manuscript is improved and ready to be published in MDPI Molecules.
On behalf of all authors, yours faithfully,
Jakub Hrubý and Petr Neugebauer